# In Silico Inference of Synthetic Cytotoxic Interactions from Paclitaxel Responses

**DOI:** 10.3390/ijms22031097

**Published:** 2021-01-22

**Authors:** Jeong Hoon Lee, Kye Hwa Lee, Ju Han Kim

**Affiliations:** 1Seoul National University Biomedical Informatics (SNUBI), Division of Biomedical Informatics, Seoul National University College of Medicine, Seoul 110799, Korea; sosal@snu.ac.kr; 2Asan Medical Center, University of Ulsan College of Medicine, Seoul 05505, Korea

**Keywords:** synthetic cytotoxicity, conditional synthetic lethality, paclitaxel, chemotherapy response, urogenital cancer

## Abstract

To exploit negatively interacting pairs of cancer somatic mutations in chemotherapy responses or synthetic cytotoxicity (SC), we systematically determined mutational pairs that had significantly lower paclitaxel half maximal inhibitory concentration (IC_50_) values. We evaluated 407 cell lines with somatic mutation profiles and estimated their copy number and drug-inhibitory concentrations in Genomics of Drug Sensitivity in Cancer (GDSC) database. The SC effect of 142 mutated gene pairs on response to paclitaxel was successfully cross-validated using human cancer datasets for urogenital cancers available in The Cancer Genome Atlas (TCGA) database. We further analyzed the cumulative effect of increasing SC pair numbers on the TP53 tumor suppressor gene. Patients with TCGA bladder and urogenital cancer exhibited improved cancer survival rates as the number of disrupted SC partners (i.e., SYNE2, SON, and/or PRY) of TP53 increased. The prognostic effect of SC burden on response to paclitaxel treatment could be differentiated from response to other cytotoxic drugs. Thus, the concept of pairwise SC may aid the identification of novel therapeutic and prognostic targets.

## 1. Introduction

Despite the presence of targeted therapies and increasing number of genomic biomarkers, cytotoxic chemotherapy, which damages cells and causes rapid cell death, remains the gold standard for most treatment of most cancers [1,2,3,4]. Paclitaxel, one of the most commonly used cytotoxic agents, is a mitotic inhibitor used for chemotherapeutic treatment of various cancers [5]. In recent years, several attempts have been made to identify biomarkers that affect paclitaxel responses in cancer cell lines and cancer patients using genetic profiles such as mRNA expression and exome sequencing [6,7,8]. Unlike target-specific drugs, cytotoxic drugs result in highly variable patient outcomes, making the prediction of responsiveness challenging using genomic profiles, which can otherwise provide some insights into nonspecific antiproliferative or cytotoxic effects [9].

Currently, several anticancer therapies exploit somatic mutations and oncogene overexpression, independent of tumor dependence on specific oncogenic pathways for survival [10,11,12,13]. Although oncogene-targeting inhibitors are effective for some cancer patients, not all cancer cells express these targets. As another treatment option, synthetic lethal approach targets one of the negative genetic interactions in the second site, which functionally disrupts both the genes simultaneously, leading to cancer cell death [10,11]. Negative genetic interaction-based approaches can be expanded for using chemotherapeutic agents based on response data such as the half-maximal inhibitory concentration (IC_50_) or the half-maximal effective concentration (EC_50_). Synthetic cytotoxicity (SC), a conditional synthetic lethal interaction, increases the cytotoxicity of chemotherapeutic agents when functions of specific gene pairs are disrupted simultaneously. SC does not occur when only one gene is disrupted. Interestingly, Li Xuesong et al. (2014) performed a series of yeast experiments, including synthetic genetic and plate assays, to show that digenic disruption due to the *TEL1*/*ATM* mutation leads to SC with camptothecin, a topoisomerase I inhibitor [14].

Many computational approaches that exploit the theory of synthetic lethality have been used to identify new therapeutic targets [12,15,16]. For example, Jang et al. used deep learning modeling to predict lethality based on an RNA regulatory network from in vitro screening data [17]. Moreover, computationally inferred candidate synthetic lethal pairs from various algorithms have already been organized, and a database has been constructed [18]. However, to the best of our knowledge, none of the computational approaches have taken advantage of the theory of SC that enhances chemotherapeutic drug responses. The highly variable nature of chemotherapy responses prevents prediction of personalized responses. Therefore, we hypothesize that in silico methods can identify SC pairs to aid the development of novel anticancer therapies.

Here, we focused on the chemotherapeutic agent paclitaxel, and using Genomics of Drug Sensitivity in Cancer (GDSC) cell line data, we identified SC mutational pairs that increase the anticancer effect of paclitaxel via conditional synthetic lethality. Moreover, we validated these findings using The Cancer Genome Atlas (TCGA) genome profiles and clinical data. Among many identified candidate SC pairs, we tested the SC partner genes of *TP53*, an important tumor suppressor gene with a high frequency of somatic mutation, to validate SC and the prognostic effects in patients with bladder urothelial and uterine corpus endometrial carcinoma included in TCGA.

## 2. Results

### 2.1. SC Network of Paclitaxel

We analyzed 407 GDSC cancer cell lines to identify somatic mutation profiles, copy number estimations, and IC_50_ values for paclitaxel [19]. This led to the identification of 142 SC pairs of mutated genes (consisting of 95 genes) by integrating the mutational and drug sensitivity profiles (Figure 1). A paclitaxel SC network was created by defining genes as nodes and positive SC interactions as edges. The SC network consisted of three main subnetworks and two singleton SC pairs with an average degree of 2.989 (Figure 2). The size of a node represents a network degree. Of the 95 genes, *AHNAK2* exhibited the highest degree of 13, followed by *PLEC* and *ANKRD30A*, both with a degree of 12.

### 2.2. SC Gene Pairs Were Enriched for Cell Death and Chemical Responses

To investigate the characteristics of these SC interactions, functional enrichment analysis was performed for the SC network genes with Gene Annotation Tool to Help Explain Relationships (GATHER) network inference. Figure 3 illustrates the result of functional enrichment analysis using Gene Ontology (GO) and the GATHER with network inference. Notably, the three clusters were significant based on enrichment analysis. The first cluster was enriched in terms related to cell proliferation and developmental processes. The second cluster was enriched in GO terms for chemical responses. The third cluster was enriched in the GO terms associated with cell death.

### 2.3. SC Burden

An SC interaction was defined as a better response to paclitaxel (i.e., lower IC_50_ value) when both genes in an interacting pair were disrupted (details in method section). By counting the number of SC pairs, we classified the 407 samples into four SC burden groups (i.e., Group 1 with no SC pair (*N* = 112), Group 2 with one to two SC pairs (*N* = 103), Group 3 with three to nine SC pairs (*N* = 86), and Group 4 with more than nine SC pairs (*N* = 106; Figure 4). As expected, the higher the SC burden, the lower the median log IC_50_ value for paclitaxel (Group 1 = −1.953, Group 2 = −3.061, Group 3 = −3.808, and Group 4 = −4.313). We determined statistical significance using the Kruskal–Wallis test (*p* < 0.001). In the four SC burden groups, the frequency rankings of SC pairs counted in patients are listed in Appendix A.

### 2.4. TP53 SC Pairs

*TP53* is a very important tumor suppressor gene with high mutational frequencies in many cancers. *TP53* forms SC pairs with three genes, spectrin repeat-containing nuclear envelope protein 2 (*SYNE2*), negative regulatory element-binding protein (*SON*), and *PRY* (Figure 2). *SYNE2* helps maintain the structural integrity of the nucleus [20]. When both *TP53* and *SYNE2* were disrupted, the response to paclitaxel was significantly better compared with that of the other three cases (*p* < 0.05, Wilcoxon test; Figure 5a). The reactome pathway database showed that related pathways for *SYNE2* included the cell cycle, mitosis, and meiosis. Additionally, *SON* promotes splicing of several cell cycle and DNA-repair transcripts containing weak splice sites [21] (Figure 5b). GO analysis revealed that *SON* may be involved in the cell cycle, mitotic cytokinesis, microtubule cytoskeleton organization, mRNA processing, and RNA splicing. *PRY* forms a cytotoxic pair with *TP53*, however, no biological process terms or pathways are known. However, this gene was excluded from subsequent analysis because *PRY* gene disruption was not observed in TCGA patients and was located on the Y chromosome (Figure 5c). Additionally, compared to the 142 SC gene pairs, the *PRY*–*TP53* pair had the least statistical significance.

### 2.5. Survival Analysis of Prognostic Subgroups of TP53 SC Network

Improved response to a chemotherapeutic agent is believed to aid the therapeutic effect in patients. Thus, to validate the effect of SC in the *TP53* network, we considered the urogenital system in the TCGA database, including bladder urothelial (*N* = 406) and uterine corpus endometrial (*N* = 545) carcinomas, which are commonly treated with paclitaxel chemotherapy [22]. Clinical characteristics in both cancer patient groups are listed in Table 1. We hypothesized that SC pairs and their increased burden would improve patient prognoses. We constructed a gene disruption matrix using somatic mutation and copy number estimation data available from TCGA. To confirm whether the accumulation of SC pairs in *TP53* subnetworks was related to patient prognosis, we divided the patients into two groups according to the presence or absence of *TP53* disruption. We only analyzed the SC interactions of *SYNE2* and *SON* with *TP53*, as no *PRY* mutations were identified in the TCGA database.

The survival effect (overall survival) was analyzed according to the presence or absence of individual co-mutations to examine gene-induced synergistic effects in combination with *TP53* on individual patient prognosis (Figure 6). Patients with bladder urothelial carcinoma were divided into four groups according to the mutations in *SYNE2* and *TP53*. There were 48 patients with mutations in *SYNE2* and *TP53*, 88 patients with mutations in *SYNE2*, 150 patients with mutations in *TP53*, and 184 patients with wild-type versions of both genes (Figure 6a). Cox regression with Firth’s penalized likelihood method was used for patients with mutations in *SYNE2* and *TP53*, as this was the reference group. Compared with the group with mutations in *SYNE2* and *TP53*, the hazard ratio and *p*-value of wild-type, *TP53* mutant, and *SYNE2* mutant groups were 2.40 and 0.047, 2.67 and 0.021, and 2.68 and 0.092, respectively. In the *TP53* and *SON* pair, there were no significant differences between each group, however, the hazard ratio of each group was greater than 1. The same method was applied to patients with uterine corpus endometrial carcinoma. When the *TP53* and *SYNE2* mutant groups were referenced, the *TP53* group had a *p* value of 0.094 and a hazard ratio of 2.70. The *SYNE2* and wild-type groups were not significantly different, however, the hazard ratio was greater than 1. With the *SON* and *TP53* mutant group as the reference, the hazard ratios of the wild-type group were 4.51 (*p* = 0.184) and 8.43 (*p* = 0.033), respectively, for the *TP53* mutant group, and 9.54 (*p* = 0.041) for the *SYNE2* mutant group.

Figure 7 illustrates the Kaplan–Meier curves for the patients with bladder urothelial (Figure 7a,b) and uterine corpus endometrial (Figure 7c,d) carcinoma, according to the cumulative disruption of *SYNE2* and *SON* with (Figure 7a,c) and without (Figure 7b,d) *TP53* mutation. Red lines represent wild-type *SYNE2* and *SON*, green lines indicate a mutation in one of the genes, and blue lines indicate disruption of both genes. While the *TP53* mutations (Figure 7a,c) were noted in both cancer types, accumulation of *SYNE2* and *SON* SC pairs showed better prognosis according to the results of Cox regression with Firth’s penalized likelihood method (*p* < 0.05). However, patients without the *TP53* mutation (Figure 7b,d) did not show any difference in prognosis due to cumulative disruption of *SYNE2* and *SON* (*p* > 0.05). There were no differences in cumulative disruption of *SYNE2* and *SON* in the *TP53* nondisrupted group. The number of patients analyzed over time are listed in the risk table under the Kaplan–Meier graph.

To confirm the statistical significance of the SC burden with the pathologic stage, multivariate Cox proportional hazard analysis was performed (Appendix A). Patients without TP53 mutations did not show statistically significant results for survival analysis, regardless of stage significance. Only TP53 mutant patients with an SC burden consistent with the univariate analysis showed significant results for survival analysis. In TP53 mutant bladder cancer, the *p* value of the SC burden was 0.031 and the hazard ratio was 0.397, while in TP53 mutant uterine cancer, the *p* value of the SC burden was 0.035 and the hazard ratio was 0.237.

### 2.6. Robustness of Synthetic Cytotoxic Pairs in Chemotherapy Agents

To assess the robustness of the synthetic cytotoxic pairs to paclitaxel, we tested whether the responses to other drugs were significantly different according to the pair burden. A total of 142 SC pairs were divided into four groups according to the burden, and the distribution of cytotoxic agent response was confirmed. Differences in the IC_50_ values according to the burden were visualized as boxplots for eight drugs: bleomycin, docetaxel, doxorubicin, epothilone B, etoposide, gemcitabine, pyrimethamine, and vinorelbine (Figure 8). Differences in IC_50_ values are indicated by the *p* value for the Wilcoxon test above the boxplot. Our results suggest that the paclitaxel SC pairs can help distinguish the response of other cytotoxic agents.

## 3. Discussion

Here, we propose a method for the identification of SC pairs that increase the sensitivity to chemotherapeutic agents, when functions of two genes are disrupted simultaneously. Unlike target-specific drugs, cytotoxic drugs result in highly heterogeneous responses making it difficult to predict patients’ responsiveness [1]. Machine learning has been used to predict drug responses based on genetic profiles [23,24,25,26]. However, most drugs with good prediction performance are target-specific, while most drugs with poor prediction performance are cytotoxic [9]. Additionally, the identification of biomarkers for predicting responses to cytotoxic agents that cause mutations or copy number changes in the GDSC database has been challenging [27]. As part of our investigation of negative genetic interactions, we proposed a methodology for inferring SC from a genetic profile to deduce gene pairs that can distinguish responses to a specific cytotoxic agent. The synthetic cytotoxic pair we identified in this study showed a prognostic value in the real patient TCGA database. Moreover, we showed that SC against paclitaxel can robustly differentiate responses to other cytotoxic agents.

Cytotoxic agents remain widely used despite the existence of various cancer therapeutic strategies such as targeted therapies and immunotherapy [23]. Cytotoxic agents are inevitably prescribed if targeted therapy is unavailable or if a patient has advanced-stage cancer [1]. Additionally, due to unpredictable therapeutic effects, most cytotoxic agents are often used in combination therapies [28,29]. In selecting a combinatory cytotoxic agent, our method can be utilized as an alternative method of personalized medicine based on an individual’s genetic profile. Selection of an anticancer treatment strategy according to an individual gene mutation and dose adjustment can improve patient prognoses and minimize side effects.

Several studies have used machine learning to predict the response to cancer drugs [29,30,31,32,33]. This advanced predictive model successfully predicts drug responses based on genomic data. Meanwhile, our simple methodology approach based on an existing concept of SC is also advantageous. Due to the selection of a large number of features (genes) compared to a small data sample and the difference between the cell line and the primary tumor, the machine learning for a complex predictive model (such as deep learning and XGBoost) may be prone to overfitting. However, our approach has been successful with other chemotherapy agents and been verified using patient data from TCGA.

Our SC inference method was designed to predict response to cytotoxic agents. Most cytotoxic agents target complex and heterogeneous processes such as cell cycle and mitosis [34,35]. Paclitaxel is a mitotic inhibitor that rapidly kills dividing cancer cells [5]. Notably, the synthetic cytotoxic genes identified in this study were associated with cell death and responses to chemicals, despite neither being markers for drug responses. Moreover, both *SYNE2* and *SON*, which form important SC interactions with *TP53*, are involved in the cell cycle [20,21]. Although a single defect in a gene involved in the cell cycle cannot be used to predict drug response, SC, which increases the dimension of the genetic profile, can be used to predict responses to cytotoxic agents reflecting the composition of the complex cell cycle. Studies showing that the tumor mutational burden does not predict cytotoxic chemotherapy responses further support this observation [36,37]. This pair-wise approach can help to identify novel biomarkers and improve the prediction based on a single biomarker, more so, since tumors are heterogeneous and have polyclonal drug-resistant properties [38].

The GDSC database provides a mutational profile for 1001 cell lines and IC_50_ values for 265 drugs. TCGA is also a large database that provides multiomics data along with the clinical information of real patients. We have shown that although the two databases are built for different purposes, the information available on them can be linked in a complementary manner [39]. We identified a novel biomarker using the concept of SC and the information available on the GDSC database and verified the effect on patient prognosis using the TCGA database.

The *TP53* mutation is an important biomarker for tumor recurrence, progression, and prognosis in urogenital cancer [40,41]. Although the *TP53* is an important predictive biomarker, developing an exploitative therapeutic strategy has been difficult. However, SC exploits the highly mutated *TP53* in an unconventional way that allows for the development of novel therapeutic strategies by exploiting other genes. For response to paclitaxel, both *SON* and *SYNE2* genes were found to increase the drug cytotoxicity when disrupted with the *TP53* gene. Likewise, identification of a novel negative genetic interaction based on frequently mutated genes (e.g., KRAS) may aid the development of new cancer therapeutic strategies.

However, it is important to note that differences in the cancer microenvironment may affect drug responses and thus, the available cell line data. Additionally, the whole exome sequencing pipeline of TCGA and variant calling using single nucleotide polymorphism (SNP) arrays result in differences in the coverage area. For example, the *PRY*, which forms SC with *TP53*, was not found in the TCGA database. The difference in these platforms also affects the subsequent results of the analysis. Our results were analyzed by increasing the dimension of a single genetic biomarker. Thus, genes with low mutational frequencies are unlikely to produce significant results in SC. We analyzed the prognosis of the SC network in TCGA bladder and uterine cancer, focusing on the TP53 gene due to its high frequency. However, the patients’ prognosis to paclitaxel is not related only to the different mutational pairs but is the result of a complex network and compensatory mechanisms. Rather than the burden of the SC pair, more research on advanced approaches using the network should be conducted. Therefore, we need to do more research on advanced approaches using the SC network in the future. There may also be tissue-specific properties that cannot be accounted for in mutation profiles. Given that this method is data intensive, it was not possible to analyze specific tissue types alone. Moreover, patients listed on TCGA were not exclusively prescribed paclitaxel; nevertheless, in uterine cancer, paclitaxel was the most used drug. Therefore, it cannot be guaranteed that the prognostic effect of the SC burden is to the responsiveness of paclitaxel alone. In TCGA database, paclitaxel was the most prescribed drug for uterine cancer; moreover, we confirmed the predictive power of the SC burden, which predicts the response of other chemotherapy agents.

Herein, we identified synthetic cytotoxic gene pairs that led to an increase in the cytotoxicity of the chemotherapeutic agent, paclitaxel, when the functions of two genes were disrupted simultaneously. Of these, the *TP53* subnetwork of synthetic cytotoxic pairs could differentiate among prognoses of patients with uterine, bladder, and urogenital cancer found in the TCGA. However, the paclitaxel SC pairs showed significantly different responses compared to response to other cytotoxic chemotherapies. Thus, cytotoxic drug biomarkers and SC pairs may be useful to facilitate a better-informed prescription of cytotoxic chemotherapeutic agents, associated with responses that are difficult to predict.

## 4. Materials and Methods

### 4.1. Cancer Cell Line Data

We downloaded molecular profiles of 1001 cancer cell lines from the COSMIC cell line project (https://cancer.sanger.ac.uk/cell_lines, version 83) [42], including the Affymetrix SNP6 array for somatic mutation profiles and copy number alterations preprocessed by the Caveman, Pindel, and PICNIC algorithms [43,44,45]. Only variants located in the cDNA region were used. The variants were filtered out based on the data from the NHLBI GO Exome Sequencing Project (frequency < 0.00025) and the 1000 Genomes project (frequency < 0.0014) to remove sequencing artifacts and germline variants [46]. SNPs with minor allele frequency were removed.

### 4.2. Gene Disruption in Cancer Cell Line Project

Variant Effect Predictor (VEP) provided by Ensembl was used to annotate the SIFT score and consequence information for each variant [47,48]. Copy number estimation was derived from the PICNIC algorithm. Homozygous gene deletions were defined as dysfunctional. For drug response data, the natural log IC_50_ values for all paclitaxel-treated cell lines were downloaded from the GDSC (http://www.cancerrxgene.org/) [19,27]. There were 407 samples with copy number estimation, somatic mutation profiles, and IC_50_ values for paclitaxel. The log IC_50_ values ranged from −6.68 to 2.17, with a median of 3.117 ± 1.91.

### 4.3. Somatic Mutation Profiles in Primary Tumors

We downloaded the somatic mutation data from the TCGA GDC Data Portal (https://portal.gdc.cancer.gov/) [22]. Variant calls from the MuTect2 pipeline were used in this study [24]. The information needed for further analysis of the somatic mutations was annotated using VEP. We only focused on somatic mutations in TCGA tumor-normal matched samples in urogenital, bladder urothelial, and uterine corpus endometrial carcinomas [25,26]. We used 406 and 471 patients with bladder and uterine cancer, respectively, for clinical variables and somatic mutation profiles. In accordance with the NCCN guidelines, patients with bladder and uterine cancer were treated with paclitaxel.

### 4.4. Building Binary Disruption Gene Matrix

The somatic mutation profile and copy number alterations were used to determine the disruption of the gene. A gene with intolerant missense mutations (SIFT < 0.05) or loss of function (nonsense, frameshift, start loss, or splice site region mutation) was classified as a disrupted gene. From the copy number alteration data, when gene deletions were homozygous, they were defined as disrupted genes. Finally, a disruption binary matrix composed of genes and samples was constructed.

### 4.5. Identification of Synthetic Cytotoxic Interactions

The workflow scheme for our analysis process is illustrated in Figure 1. For all possible combinations of gene pairs, the Wilcoxon test was used to evaluate the differences in paclitaxel IC_50_ values. We divided the samples into four groups according to the presence of loss of function mutations for two genes (i.e., one group with loss of function in both genes, two groups with loss of function in each gene independently, and a group with wild-type versions of both genes). Tests were performed on genes disrupted in more than 5% of samples. The test was performed only if there were at least 2.5% of the samples in each group. For each gene pair, the Wilcoxon test was performed three times for the group in which both genes were disrupted and in the other three groups. For the gene pair, a synthetic cytotoxic pair was defined when the group with a disruption in two genes showed significantly lower IC_50_ values than the other groups, based on a Wilcoxon test and *p* value of 0.05.

### 4.6. Synthetic Cytotoxic Network for Paclitaxel

Genes defined as SC pairs were analyzed using the igraph software package [49]. The synthetic cytotoxic network was visualized using the Gephi program [50]. For network visualization, each gene node was colored by the modularity class deduced by the Louvain method community detection algorithm [51]. Since *TP53* is an important tumor suppressor gene that is highly mutated in various cancer types, we prioritized analyzing the genes that form SC with *TP53*.

### 4.7. Functional Enrichment Test

To confirm the function of the inferred gene pairs, we used the gene ontological biological process term and the Kyoto Encyclopedia of Genes and Genomes (KEGG) pathway [52,53]. GATHER is an online tool for functional enrichment analysis that interprets through enriched functions in input genes (https://changlab.uth.tmc.edu/gather/gather.py) [54]. This tool integrates gene function, ontology, and pathways to provide an interpretation of genomic data analysis results with network inference. Enrichment analysis was performed with the latest version of GO and KEGG pathway using R package; Bioconductor’s RDAVIDWebService version 3.12 was used for genes included in the SC network with GATHER network inference genes [55].

### 4.8. Prognostic Effect of SC in TCGA Datasets

To validate SC effects for gene pairs computationally inferred from the cell line database derived from GDSC, we confirmed the prognostic effect in the real patient database from TCGA for the SC pairs that improved the response to paclitaxel. It was difficult to test the significance of genes with low mutation frequencies. Therefore, we only performed survival analysis for *TP53* synthetic cytotoxic subnetwork gene pairs. Patients were divided into two groups, with and without *TP53* mutations. For each group, patients were subdivided according to the burden of the mutation forming SC with *TP53*. Finally, Cox regression with Firth’s penalized likelihood method was used based on the burden of the mutation, and the hazard ratio of the burden was compared between the *TP53* mutant and nonmutant groups to confirm the prognostic effect of the SC pair [56].

## Figures and Tables

**Figure 1 ijms-22-01097-f001:**
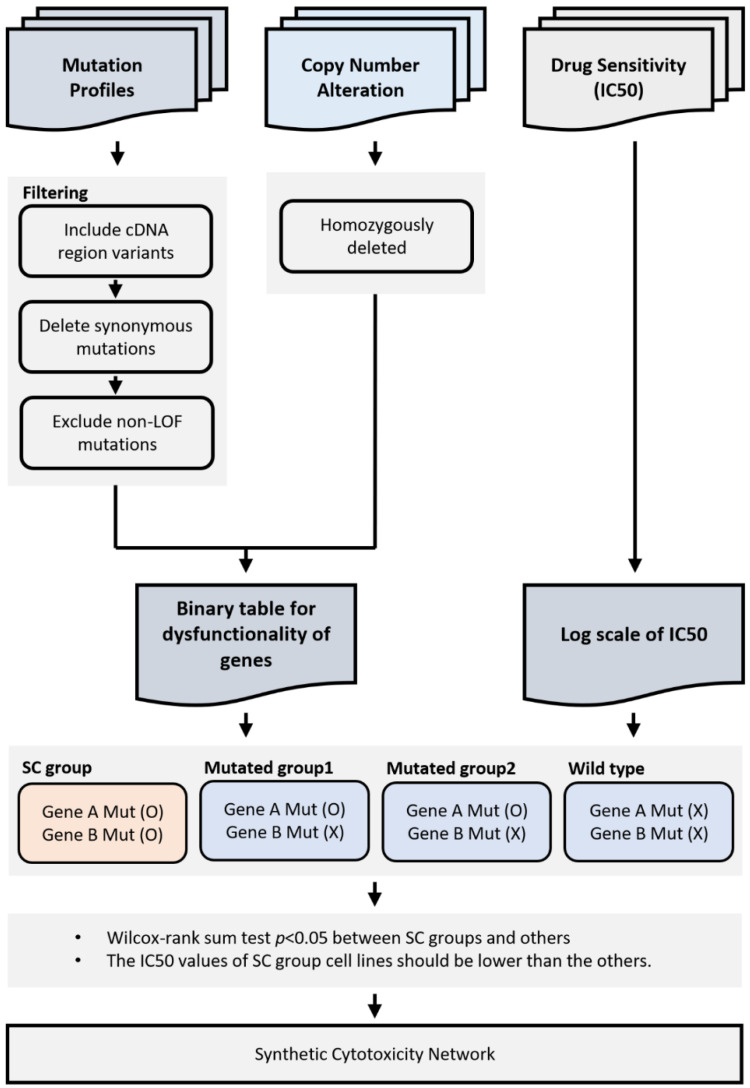
Workflow scheme for inferring synthetic cytotoxicity. All cancer cell lines were divided into four groups and defined based on the disruption of two genes. A synthetic cytotoxic pair is defined when both disrupted genes have significantly lower IC_50_ than those in the other three groups.

**Figure 2 ijms-22-01097-f002:**
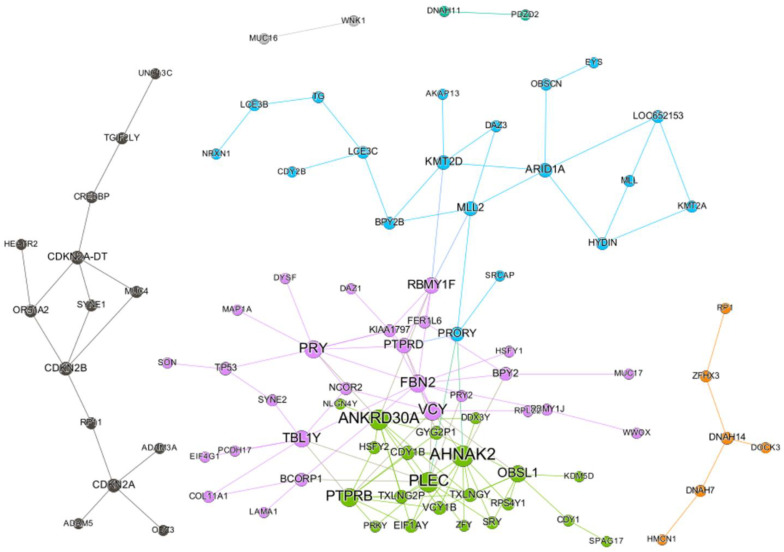
Inferred synthetic cytotoxic interaction network of paclitaxel.

**Figure 3 ijms-22-01097-f003:**
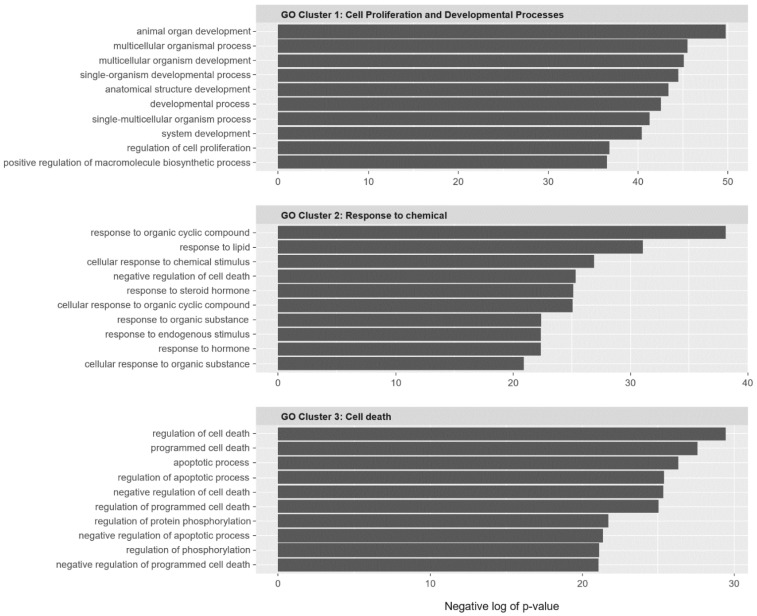
Functional enrichment analysis of the Kyoto Encyclopedia of Genes and Genomes (KEGG) pathway and gene ontology for 95 genes involved in synthetic cytotoxic interactions. These genes were significantly enriched in terms related to cell death and response to chemical and biological processes.

**Figure 4 ijms-22-01097-f004:**
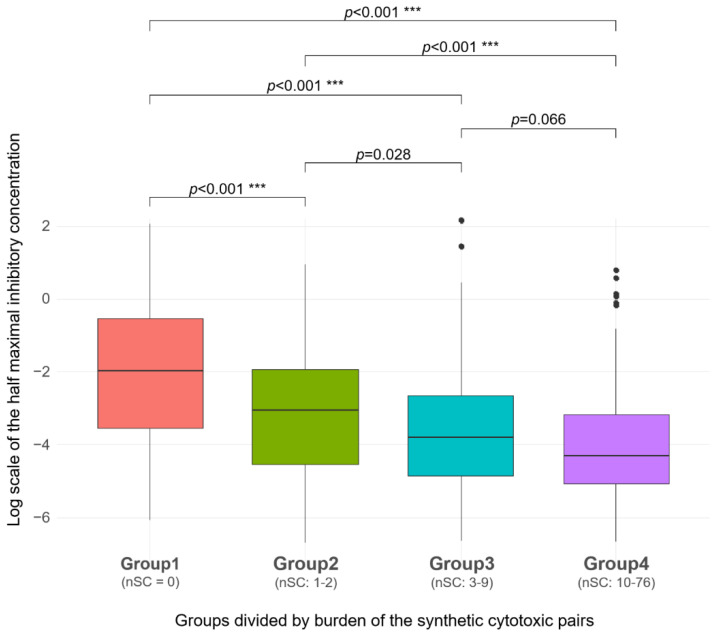
In the four groups divided according to the number of synthetic cytotoxic pairs, groups with more pairs had lower IC_50_ values. There were 112, 103, 86, and 106 samples in Group 1 to Group 4, respectively.

**Figure 5 ijms-22-01097-f005:**
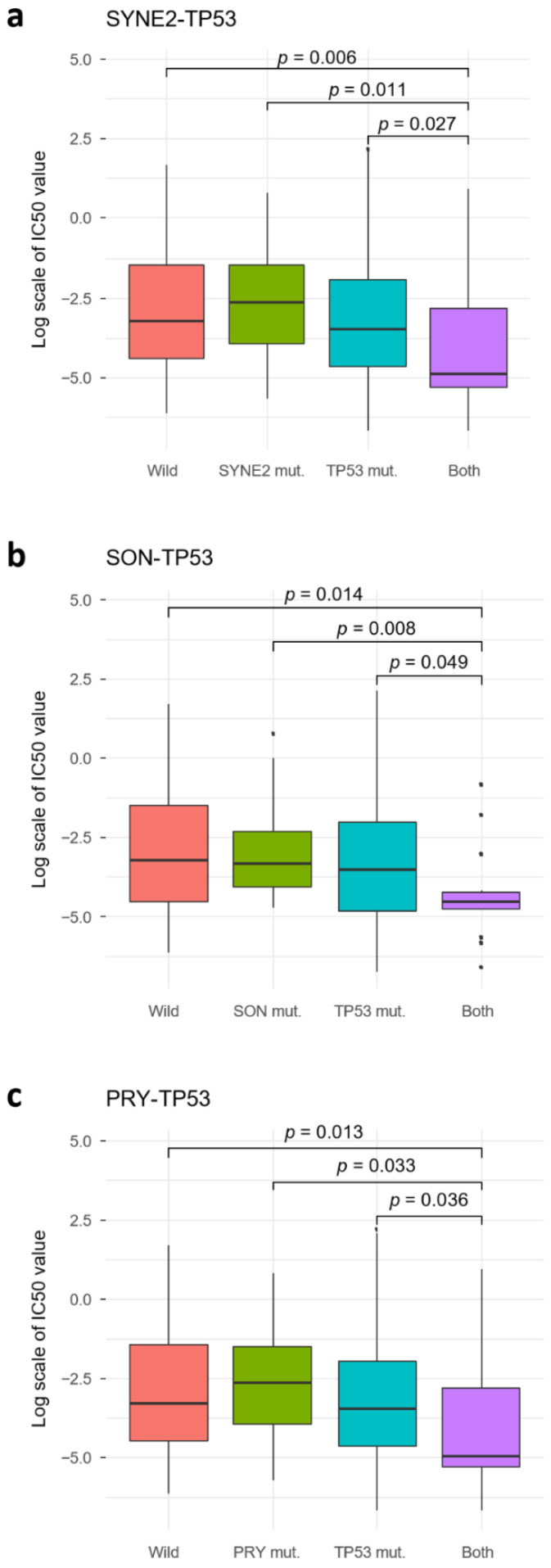
(**a–c**) Three genes that exert a synergistic effect on TP53 responses to paclitaxel. When the SYNE2, SON, and PRY genes digenically disrupt *TP53*, the IC_50_ value of paclitaxel decreases significantly.

**Figure 6 ijms-22-01097-f006:**
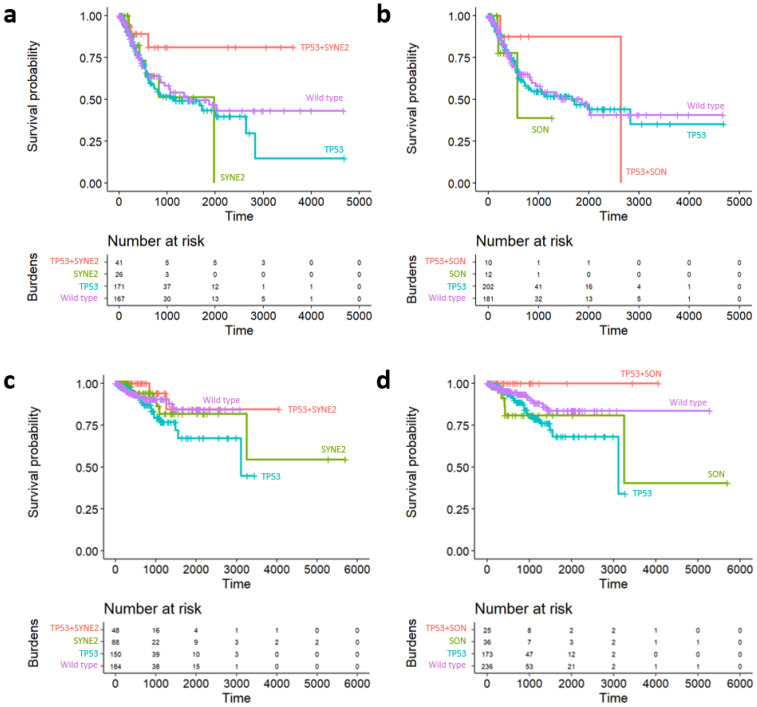
Survival curves according to the events of pair genes constituting synthetic cytotoxicity. (**a**) Patients with disruption in TP53 and SYNE2 mutations in bladder cancer. (**b**) Disruption in TP53 and SON mutations in bladder cancer. (**c**) Patients with disruption in TP53 and SYNE2 mutations in uterine cancer. (**d**) Disruption in TP53 and SON mutations in uterine cancer.

**Figure 7 ijms-22-01097-f007:**
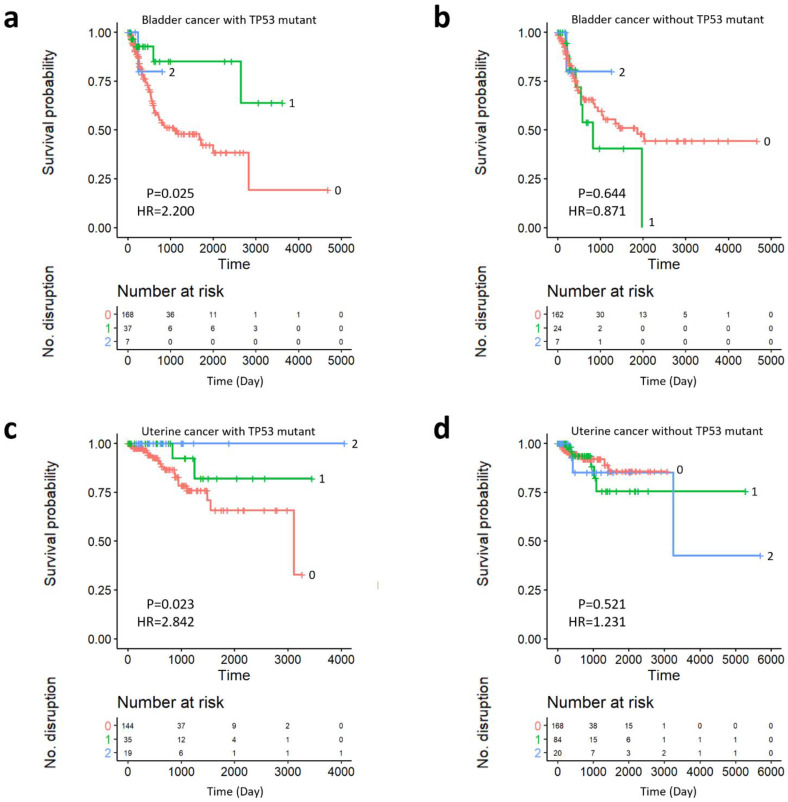
Kaplan–Meier graph for cumulative disruption of SYNE2 and SON, without the TP53 mutation. Red lines denote wild-type SYNE2 and SON, green lines represent mutation in one of the genes, and blue lines represent the disruption of both genes. (**a**) Patients with bladder cancer with a TP53 mutation. (**b**) Patients with bladder cancer without a TP53 mutation. (**c**) Patients with uterine cancer with a TP53 mutation. (**d**) Patients with uterine cancer without a TP53 mutation.

**Figure 8 ijms-22-01097-f008:**
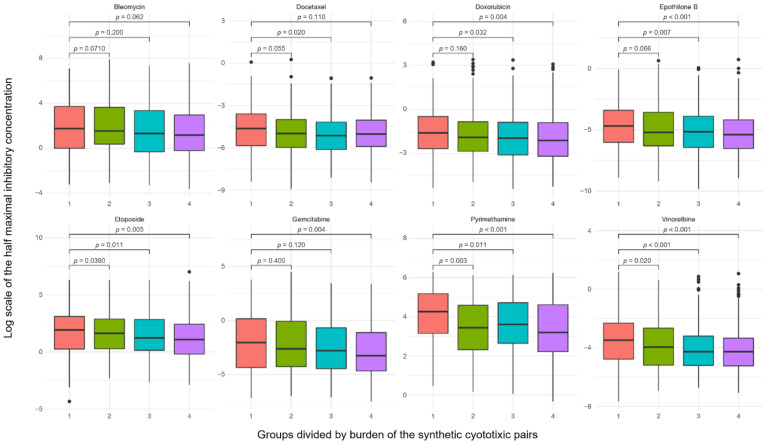
The four groups were divided according to the number of synthetic cytotoxic pairs. The burden of the synthetic cytotoxic pair can significantly differentiate the response to paclitaxel from that to other cytotoxic drugs.

**Table 1 ijms-22-01097-t001:** Patient characteristics of TCGA validation dataset, bladder urothelial carcinoma and uterine corpus endometrial carcinoma.

Clinical Variables	Bladder Cancer		Uterine Cancer	
	TP53 (+)	TP53 (−)	TP53 (+)	TP53 (−)
**N**	212	194	199	272
**Age**	63.73 (9.87)	63.01 (11.87)	61.98 (9.88)	58.31 (11.24)
**Survival**				
Alive	151	145	178	252
Dead	61	49	21	45
**Gender**				
Male	153	149	0	0
Female	59	47	199	272
**Stage**				
I	0	2	98	189
II	63	67	20	25
III	74	64	63	50
IV	75	59	18	8

## Data Availability

The data analyzed in this study is available from public repositories. These data can be found here: TCGA, https://portal.gdc.cancer.gov/; GDSC, http://www.cancerrxgene.org/.

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
