# Peer review of "In Silico Inference of Synthetic Cytotoxic Interactions from Paclitaxel Responses"

_ijms, 2021, doi:10.3390/ijms22031097_

Round 1

Reviewer 1 Report

The authors investigated negatively interacting pairs of cancer somatic mutations in chemotherapy responses or synthetic cytotoxicity (SC) and systematically determined mutational pairs that significantly lower paclitaxel IC50 values in the treatment of urogenital cancers from 407 GDSC cell lines with somatic mutation profiles, copy number estimations, and drug-inhibitory  concentrations. This is an intersting topic and the results may provid some therapeutic and prognostic targets. However, there are several major concerns regarding this study:

  1. The database of GATHER was shut down In December 2016. From 2016 to 2020, there are some updated versions for databases (such as KEGG). Why did the authors choose this database? The authors should re-perform the analysis by  using updated databases.
  2. Please clarify the paired genes were included in figure 4.
  3. What is the definition of “lower IC50”?
  4. The result of figure7a and 7b are not consistent. In figure 7B, patients with a mutation in one of the genes had lower survival. But, in figure 7a, patients with triple mutation genes had lower survival than patients with double mutation genes. Please discuss this result. Besides, please explain the cumulative disruption.
  5. Paclitaxel is also recommended in the first-line setting of gastric cancer. Why did the authors exclude this cancer?

Reviewer 2 Report

Lee and colleagues proposed a computational research article aimed at elucidating the role of gene mutations in the response of paclitaxel by using a synthetic cytotoxic interaction algorithm. The first part of the manuscript aimed at assessing the SC network of paclitaxel, the SC gene pairs and the SC burden looks very interesting and answer to important issues related to the polyclonality of tumors and to the importance of the tumor mutational burden in the choice of the best effective treatments. However, the second part of the study is less interesting and could lead to important bias because of the analysis of TP53 SC pairs alone, without considering other important tumor-suppressor genes or oncogenes. Overall, the manuscript is well written, below are reported some comments that will improve the quality of the manuscript:

1) This is a too simplistic model. Indeed, the patients’ response to paclitaxel is not related only to the different mutational pairs affecting one or two genes but is the results of a complex network and compensatory mechanisms according to which when one gene is blocked by the treatment, other genes numerous other genes are activated to compensate the loss of function of the gene affected by the treatment itself or by treatment-induced mutations. Therefore, such analyses performed on chemotherapeutic agents like paclitaxel should consider a network of genes instead of single gene pairs. Please argue better this issue in the Discussion section;

2) Have the authors stratified TCGA patients also according to tumor stage for the Kaplan Meier analyses? Indeed, differences in paclitaxel response may be due to tumor stage. In addition, as reported in Table 1, the distribution of TP53 mutation in bladder cancer and uterine carcinoma is different according to the tumor stage. Please, clarify;

3) In the Discussion section, the authors should discuss the importance of the tumor mutational burden in the response of chemotherapy. In addition, some information about the different cell clones within the tumor should be added. Indeed, tumor cannot be considered as a single-cell entity, but it can be considered as the results of different cell clones with a different subset of driver and passenger mutations. Please, better argue these issues. For this purpose, see:

- 10.1080/2162402X.2020.1781997

- 10.1186/s13000-020-00971-7

- 10.1016/j.molonc.2014.06.005+

4) Please provide more updated references for the first two sentences of the Introduction section. For this purpose, see:

– 10.3389/fphar.2018.01300

– 10.3390/molecules25235776

5) Please, correct some errors occurring in the manuscript.

Reviewer 3 Report

The Manuscript entitle “In silico inference of synthetic cytotoxic interactions from paclitaxel responses” is well written and well designed. The methodologies are explained in detail. The work may have a relevant impact in predicting which patients may eventually benefit from specific treatments with cytotoxic drugs. The major concern of this work is the lack of in vitro data to validate this specific methodology. In addition, the word “validation” using along the abstract and the manuscript seems to indicate that in vitro or in vivo studies or even patient samples were used to validate the in sicilo data, which was not the case.

Minor comments/suggestions:

  • Figure 5: add the “p” before the number, and include the statistic.
  • Line 156: correct “endometrial (Figure 7A, C)” to “endometrial (Figure 7C, D)”.
  • Figure 6: clarify if the survival means “overall survival”, and clarify what is the scale of the “time”.
  • Figure 7: In the graphs, identify which corresponds to bladder cancer and uterine cancer, as well as, TP53 mutation or without TP53. In addition, it would be better to include near the lines, the name of the correspondent genes (as done in Figure 6). This would help to visualize the data in the graphs.
  • Figure 8: Indicate in the figure legend or in the graphs, what corresponds each group (1 to 4). As before, add the “p” before the number, and include the statistic.
  • In materials and methods, 4.3 section: confirm that patients selected from the TCGA analysis were treated only with paclitaxel, or that patients were also treated with other drugs simultaneously (if so, this should be taken into consideration).

Reviewer 4 Report

I have several suggestion that in my opinion will improve the quality of the manuscript.

(1) Introductory section does not fully unfold the currently used techniques. Specifically, what has been already done by computational specialists in this area of research?

(2) For sake of reproducibility, authors should consider expanding Materials and Methods section by providing more specific details. For example, line 261 "SNPs with minor allele frequency were removed" - what was the cut-off for frequencies? 

(3) Discussion has several mismatches with data shown in Results. Authors should carefully revise all statements so the results and discussion match each other.

Author Response

I have several suggestion that in my opinion will improve the quality of the manuscript.

(1) Introductory section does not fully unfold the currently used techniques. Specifically, what has been already done by computational specialists in this area of research?

Response: Thanks for your detailed review. Following your advice, we have revised the 3rd paragraph of the Introduction section as follows with a new latest research reference.

Many computational approaches that exploit the theory of synthetic lethality have been used to identify new targets [12,15,16]. An example is a study of deep learning modeling that predicts lethality based on an RNA regulatory network using vitro screening data [17]. Moreover, computationally inferred candidate synthetic lethal pairs from various algorithms have already been organized, and a database has been constructed [18]. However, to the best of our knowledge, no computational approaches have taken advantage of the theory of SC that increases chemotherapeutic drug responses. The highly variable nature of cytotoxic chemotherapy responses prevent personalized response prediction. Therefore, we hypothesize that in-silico methods can be utilized to identify SC pairs that can be used to aid the development of novel much-needed anticancer therapies.

(2) For sake of reproducibility, authors should consider expanding Materials and Methods section by providing more specific details. For example, line 261 "SNPs with minor allele frequency were removed" - what was the cut-off for frequencies? 

Response: Thanks for your detailed comment. To reflect the information of cut-off for frequencies in manuscript, Materials and Methods section, 4.1 paragraph has been modified as follows. The cut-off below is the threshold officially used in the COSMIC cell line project.

The variants were then filtered out based on the data from the NHLBI GO Exome Sequencing Project (frequency < 0.00025) and the 1000 Genomes project (frequency < 0.0014) to remove sequencing artifacts and germline variants [35]. SNPs with minor allele frequency were removed.

(3) Discussion has several mismatches with data shown in Results. Authors should carefully revise all statements so the results and discussion match each other.

Response: Thanks for your comment. We have modified the contents of the Discussion section and added a new paragraph.

Round 2

Reviewer 1 Report

The authors have revised all the concerns.

Author Response

The authors have revised all the concerns.  

Response: Thank you so much for your advance comments.

Reviewer 2 Report

The authors well addressed almost all of my comments. There are still some aspects that should be clarified regarding comment 1 of my first report: 

1) This is a too simplistic model. Indeed, the patients’ response to paclitaxel is not related only to the different mutational pairs affecting one or two genes but is the results of a complex network and compensatory mechanisms according to which when one gene is blocked by the treatment, other genes numerous other genes are activated to compensate the loss of function of the gene affected by the treatment itself or by treatment-induced mutations. Therefore, such analyses performed on chemotherapeutic agents like paclitaxel should consider a network of genes instead of single gene pairs. Please argue better this issue in the Discussion section;

In particular, the authors should better argue the importance of analyzing more than two genes to completely evaluate the response to cytotoxic agents. Please, add in the conclusive remarks the limitation of the study, that is the study of the mutational pairs affecting only one or two genes with particular reference to all the hallmark mutations of cancer and the polyclonality of tumors.

Author Response

The authors well addressed almost all of my comments. There are still some aspects that should be clarified regarding comment 1 of my first report: 

1) This is a too simplistic model. Indeed, the patients’ response to paclitaxel is not related only to the different mutational pairs affecting one or two genes but is the results of a complex network and compensatory mechanisms according to which when one gene is blocked by the treatment, other genes numerous other genes are activated to compensate the loss of function of the gene affected by the treatment itself or by treatment-induced mutations. Therefore, such analyses performed on chemotherapeutic agents like paclitaxel should consider a network of genes instead of single gene pairs. Please argue better this issue in the Discussion section;

In particular, the authors should better argue the importance of analyzing more than two genes to completely evaluate the response to cytotoxic agents. Please, add in the conclusive remarks the limitation of the study, that is the study of the mutational pairs affecting only one or two genes with particular reference to all the hallmark mutations of cancer and the polyclonality of tumors.

Response: Thank you for your comment. As you advised, analyzing the prognosis of the SC network and focusing on the TP53 gene in TCGA bladder and uterine cancer is a distinct limitation. Since the TP53 gene is a highly frequently found mutation, we focused on this analysis. However, other SC gene pairs are also worth for studying. Therefore, we have added limitations and future work to the discussion section as follows (The added part is displayed in bold).

However, it is important to note that differences in the cancer microenvironment may affect drug responses and thus, the available cell line data. Additionally, the whole exome sequencing pipeline of TCGA and variant calling using single nucleotide polymorphism (SNP) arrays result in differences in the coverage area. For example, the PRY, which forms SC with TP53, was not found in the TCGA database. The difference in these platforms also affects the subsequent results of the analysis. Our results were analyzed by increasing the dimension of a single genetic biomarker. Thus, genes with low mutational frequencies are unlikely to produce significant results in SC. We analyzed the prognosis of the SC network in TCGA bladder and uterine cancer, focusing on the TP53 gene due to its high frequency. However, the patients’ prognosis to paclitaxel is not related only to the different mutational pairs but is the results of a complex network and compensatory mechanisms. Rather than the burden of the SC pair, more research on advanced approaches using the network should be conducted. Therefore, we need to do more research on advanced approaches using SC network in the future. There may also be tissue-specific properties that cannot be accounted for in mutation profiles. Given that this method is data intensive, it was not possible to analyze specific tissue types alone. Moreover, patients listed on TCGA were not exclusively prescribed paclitaxel; nevertheless, in uterine cancer, paclitaxel was the most used drug. Therefore, it cannot be guaranteed that the prognostic effect of the SC burden is to the responsiveness of paclitaxel alone. In TCGA database, paclitaxel was the most prescribed drug for uterine cancer; moreover, we confirmed the predictive power of the SC burden, which predicts the response of other chemotherapy agents.

Reviewer 4 Report

Authors addressed raised concerns.

Author Response

Authors addressed raised concerns.

Response: Thank you so much for your comment.